# Enhancement-constrained acceleration: A robust reconstruction framework in breast DCE-MRI

**Ty O. Easley**[1], **Zhen Ren**[2], **Byol Kim**[3], **Gregory S. Karczmar**[2], **Rina F. Barber**[4], **Federico D. Pineda**[2]*

**1** McKelvey School of Engineering, Washington University in St. Louis, St. Louis, Missouri, United States of America, **2** Department of Radiology, University of Chicago, Chicago, Illinois, United States of America, **3** Department of Biostatistics at the University of Washington, Seattle, Washington, United States of America, **4** Department of Statistics, University of Chicago, Chicago, Illinois, United States of America

* fdp@uchicago.edu

**Data Availability Statement:** All code will be available at https://github.com/tyo8/ECA_Demo A minimal data set, including run and error analysis scripts, are available at:

## Abstract

In patients with dense breasts or at high risk of breast cancer, dynamic contrast enhanced MRI (DCE-MRI) is a highly sensitive diagnostic tool. However, its specificity is highly variable and sometimes low; quantitative measurements of contrast uptake parameters may improve specificity and mitigate this issue. To improve diagnostic accuracy, data need to be captured at high spatial and temporal resolution. While many methods exist to accelerate MRI temporal resolution, not all are optimized to capture breast DCE-MRI dynamics. We propose a novel, flexible, and powerful framework for the reconstruction of highly-under-sampled DCE-MRI data: enhancement-constrained acceleration (ECA). Enhancement-constrained acceleration uses an assumption of smooth enhancement at small time-scale to estimate points of smooth enhancement curves in small time intervals at each voxel. This method is tested *in silico* with physiologically realistic virtual phantoms, simulating state-of-the-art ultrafast acquisitions at 3.5s temporal resolution reconstructed at 0.25s temporal resolution (demo code available here). Virtual phantoms were developed from real patient data and parametrized in continuous time with arterial input function (AIF) models and lesion enhancement functions. Enhancement-constrained acceleration was compared to standard ultrafast reconstruction in estimating the bolus arrival time and initial slope of enhancement from reconstructed images. We found that the ECA method reconstructed images at 0.25s temporal resolution with no significant loss in image fidelity, a 4x reduction in the error of bolus arrival time estimation in lesions ($p < 0.01$) and 11x error reduction in blood vessels ($p < 0.01$). Our results suggest that ECA is a powerful and versatile tool for breast DCE-MRI.

## Introduction

Dynamic contrast enhanced MRI (DCE-MRI) is an important tool for the diagnosis of breast cancer. MRI detects cancers that other screening methods fail to detect [1,2]. DCE-MRI is particularly important for patients with dense breasts or at high risk for breast cancer. DCE-MRI

https://figshare.com/projects/Enhancement-Constrained_Acceleration_ECA_/120072.

**Funding:** R.F.B. was supported by the National Science Foundation via grant DMS-1654076. https://www.nsf.gov/ The work described in this manuscript was supported by a grant from the National Institutes of Health under project number 5R01CA218700-04, which was awarded to G.S.K. https://www.nih.gov/ The funders had no role in study design, data collection and analysis, decision to publish, or preparation of the manuscript.

**Competing interests:** The authors have declared that no competing interests exist.

is highly sensitive (93% [1]) to invasive cancers, and has a variable and sometimes high false positive rate. One 2016 meta-analysis puts the specificity at 71% [1], while another puts it between 78% and 94% [2]; individual studies have reported specificities as low as 37% [3]. These results suggest a need for acquisition and analysis methods that increase the diagnostic accuracy of DCE-MRI. Quantitative measurement of the parameters that describe contrast uptake kinetics offers one route to improved specificity [4,5], but accurate measurement of these parameters can prove challenging.

Clinical standard-of-care focuses on morphological analysis of DCE-MRI images, as well as evaluation of the overall kinetics of the contrast uptake and washout. As a result, clinical MRI protocols produce post-contrast-injection images at very high spatial resolution; these show patient anatomy in exquisite detail but require long scan times. In the standard-of-care setting, DCE-MRI images are acquired at temporal resolutions of 60–90 seconds [6]. These temporal resolutions are too low to accurately measure kinetic parameters, especially in early uptake, when signal changes rapidly, particularly in cancers. Findings in recent years indicate that lesion conspicuity is highest immediately after contrast uptake [7,8], so it is especially important to faithfully capture early-uptake kinetics. Other modes of analysis, even primarily morphological ones, also benefit from increased temporal resolution. Some groups have found that texture features, often used to characterize the morphology of lesions and classify them into benign and malignant subgroups, become more accurate with the inclusion of kinetic data [9–12]. Thus, high temporal resolution DCE-MRI may offer significant advantages in diagnostic accuracy.

Accurate quantitative measurement of the arterial input function (AIF) requires high temporal resolution measurements and an accurate estimate of the bolus arrival time (BAT). Henderson et al. [13] found that a temporal resolution of 1s is necessary to capture the AIF. Parker et al. [14] opt instead for a population-average AIF, which they calculate from 5s/image data. Estimates of the optimal temporal resolution for pharmacokinetic analysis vary based on the underlying assumptions used to model tissue behavior. For example, Kershaw et al. showed that the standard compartmental (AATH) model requires a temporal resolution of at least 1.5s for accurate diagnosis [15], even when a population AIF is used. In small mammals, which require small fields-of-view, researchers have been able to characterize AIFs with significantly higher sampling frequency; Yankeelov et al. [16] measured an AIF at 0.9s/image in mice, while Kershaw et al. measured an AIF at 0.44s/image in rabbits [17]. Current state-of-the-art in ultrafast breast DCE-MRI produces full 3D bilateral breast scans with temporal resolution 2.7–3.8s [4,7,18,19], well above the desired threshold of temporal resolution.

In fact, the thresholds of 1s and 1.5s offered here represent necessary conditions for only a subset of analytic approaches. Fractional-second temporal resolutions in breast DCE-MRI may allow new modes of kinetic characterization, including detailed local measurements of arterial blood flow and effects of the cardiac cycle, interstitial pressure and vessel permeability, and the initial time and early morphology of lesion enhancement. These parameters have potential as indicators of malignancy [4]. However, these approaches have not been adequately explored, since they require data with high resolution in both the spatial and temporal domains. High spatio-temporal resolution data could offer significant advances in the characterization of tumor physiology and access to biomarkers previously unavailable through established techniques. In order to characterize these vascular properties and assess their utility as malignancy biomarkers, we must first develop and validate methods for the acquisition and reconstruction of high spatio-temporal resolution DCE-MRI data.

Many groups have proposed techniques to increase temporal resolution in MRI, each with their own sets of trade-offs and optimal use cases. While many methods straddle categories, most algorithms fit approximately into one of the groups below:

*Ultrafast* methods [5,7,18,19] tend to rely on parallel imaging (and partial Fourier sampling) techniques to accelerate scans with greatly reduced coverage and/or spatial resolution. While easy to implement and well-suited to kinetic analysis, the images produced at reduced coverage/resolution do not always contain enough morphological data to be clinically interpretable.

*Parallel Imaging* techniques [20–23] make use of multiple receiver coils to acquire imaging data "in parallel," with data from each coil constraining the image reconstruction. These approaches are powerful and ubiquitous, and they can be used in tandem with many other techniques. However, they suffer from highly nonlinear artifacts at high accelerations. The impact of these artifacts on pharmacokinetic analysis is not well-characterized. They also rely heavily on coil sensitivity maps, which can be difficult to measure precisely.

*View-sharing* methodologies [7,24–30] accelerate acquisitions by sampling *k*-space with non-uniform densities, sampling low frequencies much more often than high spatial frequencies. This category includes many common acquisition sequences, including DISCO, TWIST, TRICKS, 4D-TRAK, and most keyhole methods. While these methods do an excellent job of categorizing large-scale enhancement patterns (e.g., average enhancement within a lesion), they sample different spatial frequencies at different temporal resolutions. This could create errors in quantitative analysis. When low spatial frequencies are sampled more often than high ones, it is difficult to reliably interpret the enhancement kinetics of small, sharp structures like blood vessels and the edges of lesions. These structures are critical for accurate clinical diagnosis.

*Compressed sensing* [31–36] strategies capitalize on the sparse enhancement of DCE-MRI in the early uptake phase to create $L^1$-constrained image reconstructions from very highly undersampled data. This provides high spatial *and* temporal resolution. However, because these approaches require sampling incoherence, they are susceptible to artifacts from non-uniform *k*-space sampling [37–39]. These artifacts are greatest when the signal is changing rapidly, as in the critical phase of early contrast uptake [37].

While all of these methods are powerful, they may not be well-suited to the task of precisely recovering early enhancement kinetics in breast DCE-MRI.

We propose a novel, flexible, and powerful framework for the reconstruction of highly-undersampled DCE-MRI data: enhancement-constrained acceleration (ECA). If raw *k*-space data is available from the scanner and the acquisition sequence used to produce that data is known, then *k*-space data can be re-partitioned into small intervals. The data sampled during each interval can then be used to reconstruct a new set of images with a temporal resolution equal to the interval length. We use the term "sweep time" to refer to the equivalent temporal resolution of a conventional (fully-Nyquist) Fourier-sampled scan at the same spatial resolution as the reconstructed image after SENSE and partial Fourier acceleration. In a recent study, bilateral ultrafast scans with complete Fourier sampling had "sweep times" between 3.4 and 4.1s [40]. To reflect the current state-of-the-art in conventional-Fourier ultrafast acquisitions, we simulate data with a sweep time of 3.5 seconds and reconstruct at a temporal resolution of 0.25 seconds. When the temporal intervals used for image reconstruction are small compared to the "sweep time," this reconstruction problem is (highly) underdetermined. To fully constrain the reconstruction problem, we introduce a penalty function that requires approximately smooth enhancement on the short timescale of the temporal resolution of the reconstructed image. In this setting, our reconstruction problem reduces to the minimization of a constrained quadratic penalty.

In the investigation presented here, we (1) develop physiologically realistic breast phantoms from patient data, (2) simulate a virtual scanner with custom pulse sequences to acquire data, and (3) compare the time-tagged reconstructions of phantom data to a "gold standard"

conventional Fourier-sampled ultrafast acquisition. In this proof-of-concept study, we aim to demonstrate that ECA reconstructions capture bolus motion in physiologically realistic phantoms more accurately than standard methods. Code that reproduces some of the examples discussed here is openly available at <github.com/tyo8/ECA_Demo>.

## Theory and methods

### Virtual phantoms

To provide flexible, programmable, and quantitative ground truths for reconstruction testing, virtual breast phantoms modeling realistic physiology and bolus propagation were created from patient data (Fig 1). Five (5) virtual breast phantoms were created from paired ultrafast and high-spatial resolution DCE-MRI datasets representing a range of pathologies. Cases 1, 2, and 4 were invasive ductal carcinomas (grades III, II, and III, respectively); cases 2 and 4 showed necrosis and calcifications. Case 3 was a grade II invasive lobular carcinoma and case 5 was a control showing no abnormal enhancement. Acquisition parameters for the original patient scans are shown in Table 1.

Virtual phantoms were created as a set of parameters (e.g., bolus arrival time and an uptake rate constant) and functions that call those parameters (e.g., the Parker AIF) to estimate whole-image signal at an input time point. We use these enhancement models to compute the time-evolution of the virtual phantom during the scanning process. The phantoms used in this experiment modeled noisy acquisition with signal from three different classes of sources; vessel, lesion, and background. Vessel and lesion signal components come from distinct models but a single procedure; all signal components are additive.

Vessels and lesions were segmented from patient images and perfusion maps were created from constraining flow equations with ultrafast data, using a method developed by Wu et. al

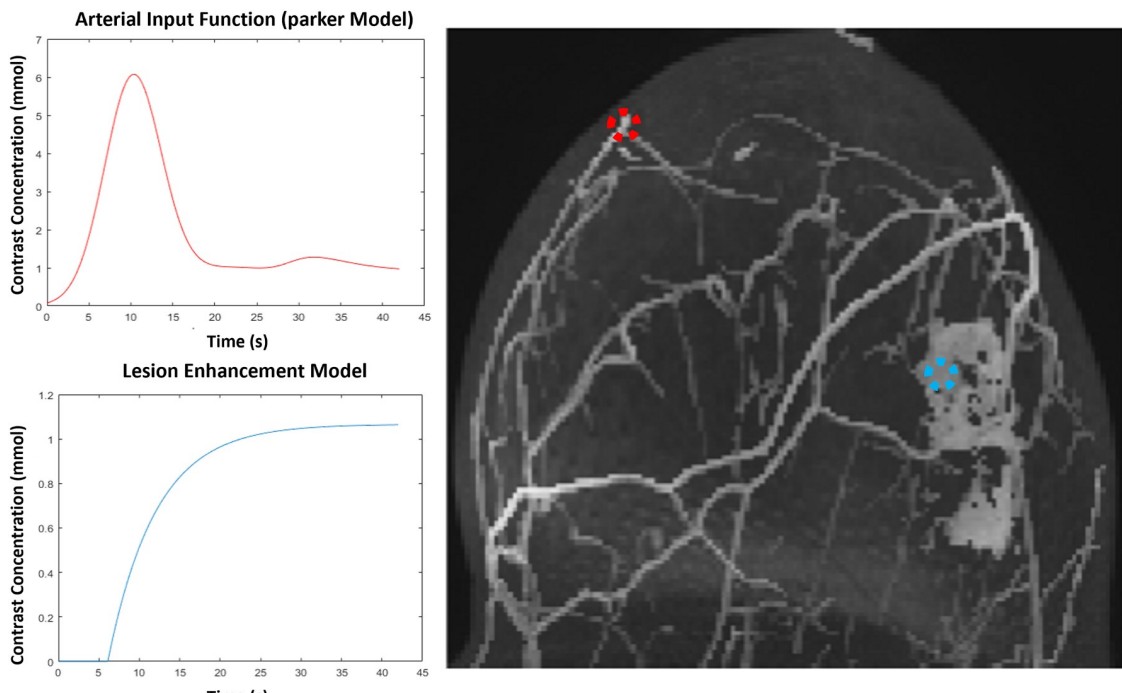

**Fig 1. A maximum-intensity projection of a sample phantom.** The red dashed circle (upper left) shows a sample voxel enhancing via the AIF; the blue dashed circle (middle right) shows a sample voxel enhancing via the EMM.

**Table 1. Scan parameters of the source MR image datasets.**

|  | HIGH-SPATIAL RESOLUTION | ULTRAFAST |
| --- | --- | --- |
| TR/TE (MS) | 4.8/2.4 | 3.2/1.6 |
| ACQUISITION VOXEL SIZE (MM$^3$) | $0.8 \times 0.8 \times 1.6$ | $1.5 \times 1.5 \times 3$ |
| TEMPORAL RESOLUTION RANGE (S) | 60–70 | 2.8–3.6 |
| FLIP ANGLE | 10˚ | 10˚ |
| FIELD OF VIEW (MM$^3$) | 300-380mm (X,Y), 180-220mm (Z) | 300-380mm (X,Y), 180-220mm (Z) |
| NUMBER OF SLICES | 120–250 | 80–100 |

[4]. First, ultrafast and high-spatial resolution image sets undergo non-rigid registration for motion correction. Next, a total 3D vasculature is segmented from high-spatial resolution image sets by a Hessian filtering process. Lesions were segmented by hand. Using the segmented lesions and vessels, ultrafast image sets were used to fit a fluid dynamics-based contrast perfusion model, which created a map of bolus arrival times in the breast. These maps of contrast perfusion times parametrized Parker et al.'s population-based AIF model [14] in vessels and a three-parameter empirically-derived enhancement model [41] in lesions. The empirical model used a truncated exponential function to model the concentration of contrast agent in lesion at initial phase (uptake only):

$$C(t) = (t \geq t_0) \cdot A \cdot \left(1 - e^{-\alpha(t-t_0)}\right)$$

where $t_0$ is the bolus arrival time in lesion voxels (s), $A$ is the upper limit of tracer concentration (mmol), and $\alpha$ is the uptake rate (s$^{-1}$). Those per-voxel parameters of the empirical model were fit from ultrafast image sets; the mean ± standard deviation of each parameter each case is listed in Table 2.

Background signal in the phantom is static except for fluctuations caused by measurement noise (noise modeling is discussed in the "Virtual Scanner" section). Motion artifacts are not directly simulated in this initial study. Each high-resolution image set contains a single pre-contrast image; these were used to compute the background signal for each phantom.

This breast phantom construction emphasizes the characteristic "sparse plus low-rank" nature of early-enhancement breast DCE-MRI [42,43] while including features that have high resolution in **both** spatial and temporal domains. In addition, this virtual phantom design is highly modular and can flexibly incorporate staggered changes to both functions and parameter sets; different morphologies, perfusion models, and enhancement functions can be swapped around with relative ease.

**Table 2. Parameter values for empirical model of tracer concentration in lesion.**

|  | $A$ | $\alpha$ (s$^{-1}$) | $t_0$ (s) |
| --- | --- | --- | --- |
| Case 1 | 1.56 ± 0.45 | 0.14 ± 0.13 | 15 ± 2.2 |
| Case 2 | 0.12 ± 0.25 | 0.019 ± 0.038 | 17 ± 7.3 |
| Case 3 | 0.84 ± 0.70 | 0.041 ± 0.033 | 20 ± 8.2 |
| Case 4 | 1.4 ± 0.38 | 0.075 ± 0.020 | 21 ± 2.8 |
| Case 5 | 1.0 ± 0.70 | 0.020 ± 0.015 | 32 ± 9.1 |

## Representation of sampling schemes

Here we clarify a notion of $k$-space sampling trajectories, which plays a role in both reconstruction and data acquisition.

A known sampling trajectory can be parametrized as a path $k_t$ through $k$-space as a function of time $t$. It specifies which Fourier coefficients are measured and the time at which each that measurement is taken. Though the present work only implements reconstruction on trajectories embedded on a Cartesian grid, any $k$-space trajectory that can be parametrized as $k_t$ can be reconstructed using the method here; any deterministic path can be simulated. Furthermore, if timestamps from a stochastic measurement are recorded during the scan (or known within the precision of reconstruction time resolution), then these data can be used for enhancement-constrained image reconstruction.

We also define a few parameters that will be useful in describing the sampling and reconstruction processes. By $N$, we denote the total number of $k$-space points acquired during a given scan. We use $T$ to represent the number of time-points in an image set. Finally, $V$ is the number of voxels in the image volume at any single time-point.

In the standard IFFT reconstruction of a Nyquist-sampled, Cartesian-acquisition $k$-space dataset, $N = VT$. On the other hand, when we form an "accelerated" (undersampled) set of images from $N$ measurements in $k$-space, we have $N < VT$. The acceleration factor is then $\alpha = \frac{VT}{N}$. In this work, we reconstruct dynamic image sets of $VT \sim 10^9$ total voxels from $N \sim 10^8$ measurements (acceleration factor $\alpha = 14$).

Though the work shown here implements a sampling scheme taken from a Cartesian grid, we emphasize that the framework presented is highly general and can be used to invert data from arbitrary sampling schemes. Furthermore, if the time course of the sampling scheme used to acquire the data is either (a) deterministic with known acquisition parameters or (b) recorded as metadata, this framework allows the retroactive reconstruction of existing data.

## Virtual scanner

A virtual scanner was developed to simulate the acquisition of data from the phantoms described above under varying $k$-space acquisition paths.

To simulate signal evolution during the acquisition window, phantom signal was recomputed for each $k$-space measurement $(k_{t_1}, \ldots, k_{t_N}$ made by the scanner. For computational efficiency, however, full re-evaluations of the virtual phantom contrast functions were only made every 50ms of scan time; signal updates between full re-evaluations were computed as linear interpolations between updates. Linearly interpolating between full updates allowed us to capture and quantify mid-scan changes in contrast dynamics, which typically occur at too fine of time-resolutions to be differentiated in standard acquisitions. These are precisely the types of signal changes we are hoping to detect. However, using linear interpolations between function evaluations implicitly encodes the assumption that contrast concentration curves are *approximately* linear in 50ms windows. and each window has sufficient SNR. While current compartmental, population, and empirical models of enhancement suggest this assumption is reasonable (see Fig 2 for a visual representation of this assumption on the Parker AIF) [41,44,45] and introduces negligible error to the models used in our phantoms, contrast perfusion curves are not sufficiently characterized to fully vet this assumption.

Virtual breast phantoms generated signal data under similar scan parameters as an ultrafast acquisition. A spoiled gradient-echo signal model was used (FA = 10˚, TR/TE = 3.2/1.6 ms, resolution = 1 mm$^3$) for each phantom acquisition. Noise was modeled as independent Gaussian distributions in $k$-space with variances computed from pre-contrast ultrafast data. To estimate interference from noise and acquisition artifacts, temporal variances were computed at

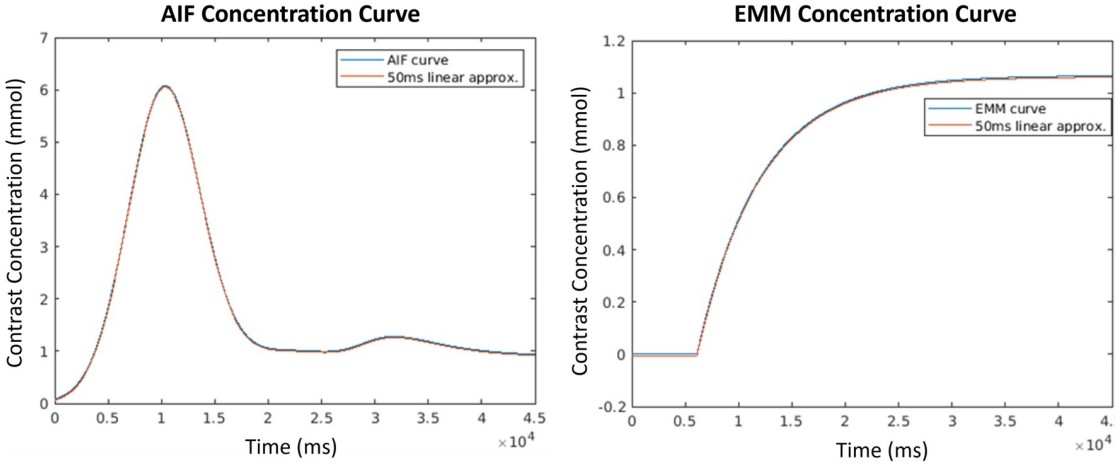

**Fig 2. Sample concentration curves paired with their 50ms-interval linear approximations.** The AIF curve (used in vessels) is shown on the left and the EMM curve (used in lesion voxels) is shown on the right.

each *k*-space point across five (5) pre-contrast ultrafast images for each phantom. Each point's temporal variance parameterized a Gaussian distribution, from which noise realizations were produced. Over all cases and 100 noise realizations, this method of noise generation produced data with an average PSNR of 37dB.

Though the virtual scanner does not have parallel imaging (e.g., SENSE or GRAPPA) [20,22] or Partial Fourier [46] implementations, path times computed from these scan parameters were appropriately scaled to match standard ultrafast time resolutions. Each scan sequence completed a Nyquist-complete *k*-space sample of *V* points every 3.5s and was later reconstructed at a temporal resolution of 0.25s. These *k*-space acquisition and reconstruction times were chosen to demonstrate the high accelerations this method can achieve (an acceleration factor of $\alpha = \frac{VT}{N} = 14$, from 3.5s to 250ms), while also allowing an investigation of the minimum time resolution necessary to fully resolve enhancement dynamics of clinical interest.

A simple but (to our knowledge) novel *k*-space trajectory was used to simulate acquisitions in this study (Fig 3). By **Un**dersampling **W**ith **R**epeated **A**dvancing **P**hase (UnWRAP), we allow our choice of reconstructed temporal resolution to inform the design of our sampling trajectory. Splitting each group of *V* acquisitions into *f* disjoint subsets, we acquire the first line of each subset before moving to the second line of the first subset: we continue in this way until all *V* acquisitions have been made. This ensures that a uniform distribution of *k*-space frequency bands determine each reconstructed image, which makes the subsequent reconstruction both (a) robust to noise and (b) sensitive to fast changes in sharp features. This sampling trajectory bears some similarities to *kt*-Blast (ref), but differs in that it enforces uniform spectral density in the sample.

The virtual scanner pipeline is summarized in Fig 4.

## IFFT reconstruction

The "standard" IFFT reconstruction is used to benchmark the ECA reconstruction. This "reconstruction" is simply the inverse fast Fourier transform applied to the *k*-space dataset output by the virtual scanner. This procedure represents the "ideal" ultrafast scan: it assumes a (spatially) Nyquist complete sample is acquired at 3.5s temporal resolution using typical ultrafast acquisition parameters (see Table 2). Because ultrafast often relies on other methods (like

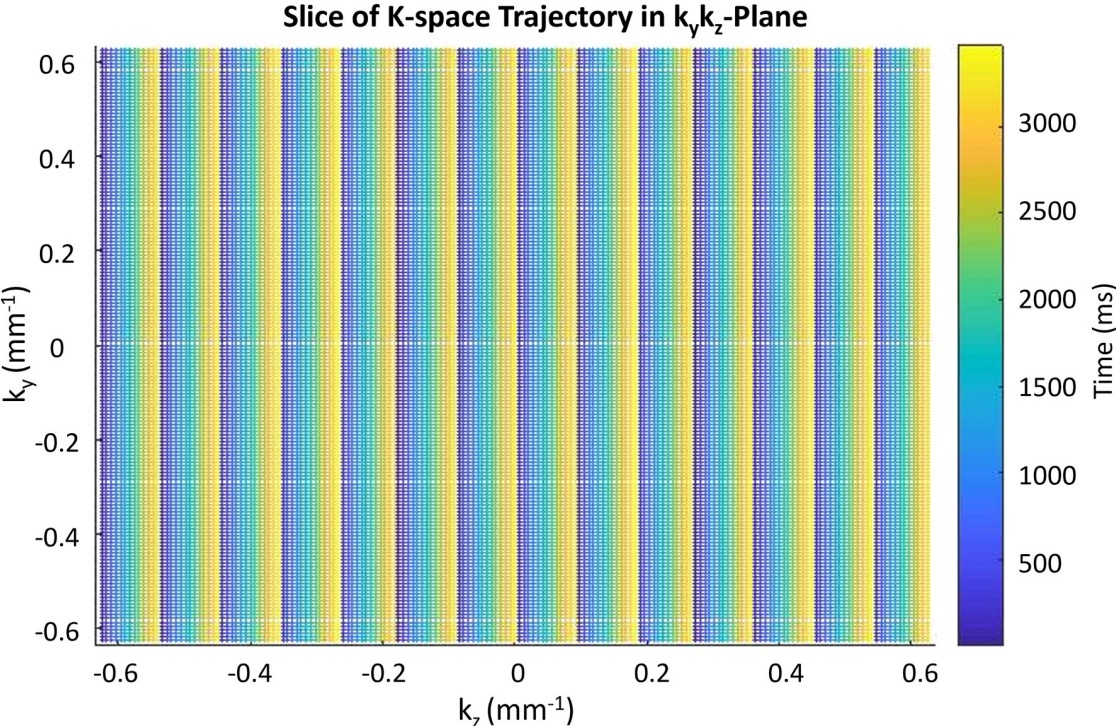

**Fig 3. Cross-section of UnWRAP sequence in the $k_y k_z$ plane.** In this scan sequence, $k$-space is divided into 14 sections, which are each separated into 14 sheaves. Each section must have one sheaf scanned before any section can have another sheaves scanned. This scheme ensures a nearly uniform distribution of high and low spatial frequencies are present in each reconstructed image, while still satisfying the (spatial) Nyquist criterion when all scan data are combined.

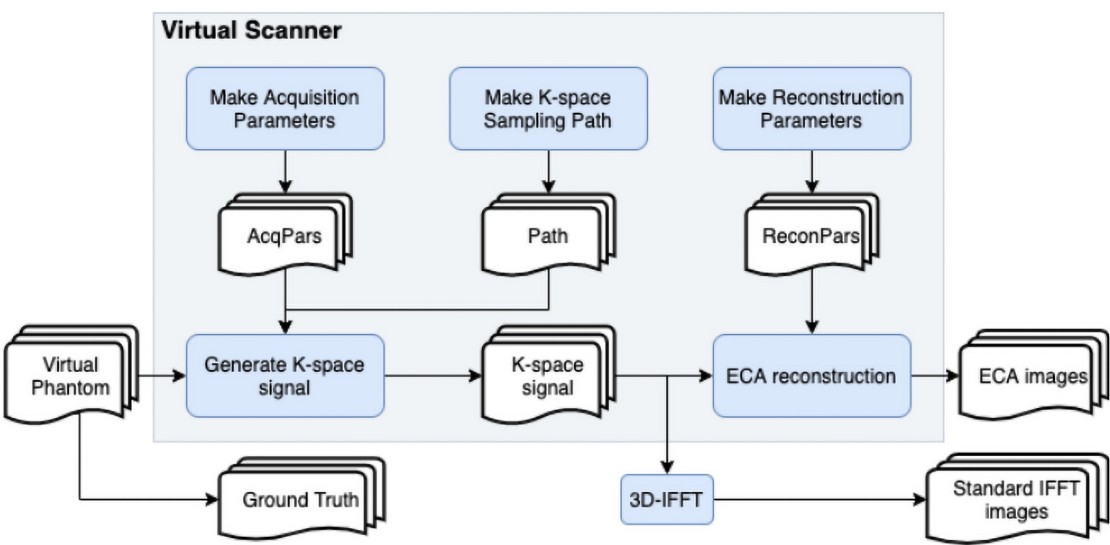

**Fig 4. A flowchart summarizing the virtual scanner pipeline.**

SENSE and Partial Fourier) to achieve such high temporal resolution [5,7,19], we are essentially benchmarking against an assumed perfect SENSE and Partial Fourier reconstruction.

## ECA reconstruction

Our enhancement-constrained acceleration (ECA) reconstruction method penalizes sharp enhancement between reconstructed time-points and requires that new images match the measured $k$-space data. The formal details of this reconstruction algorithm may be found in S1 and S2 **Appendices**, which can be summarized as follows:

1. Partition $k$-space data into time intervals of equal length; require that each measurement constrains only the image reconstructed in the time interval containing that measurement's time-tag (Fig 5).

2. Denote the reconstructed image by the timeseries $X = (X_1, \ldots, X_T)$ and its spatial Fourier transform as $\tilde{X} = \left(\tilde{X}_1, \ldots, \tilde{X}_T\right)$. Define

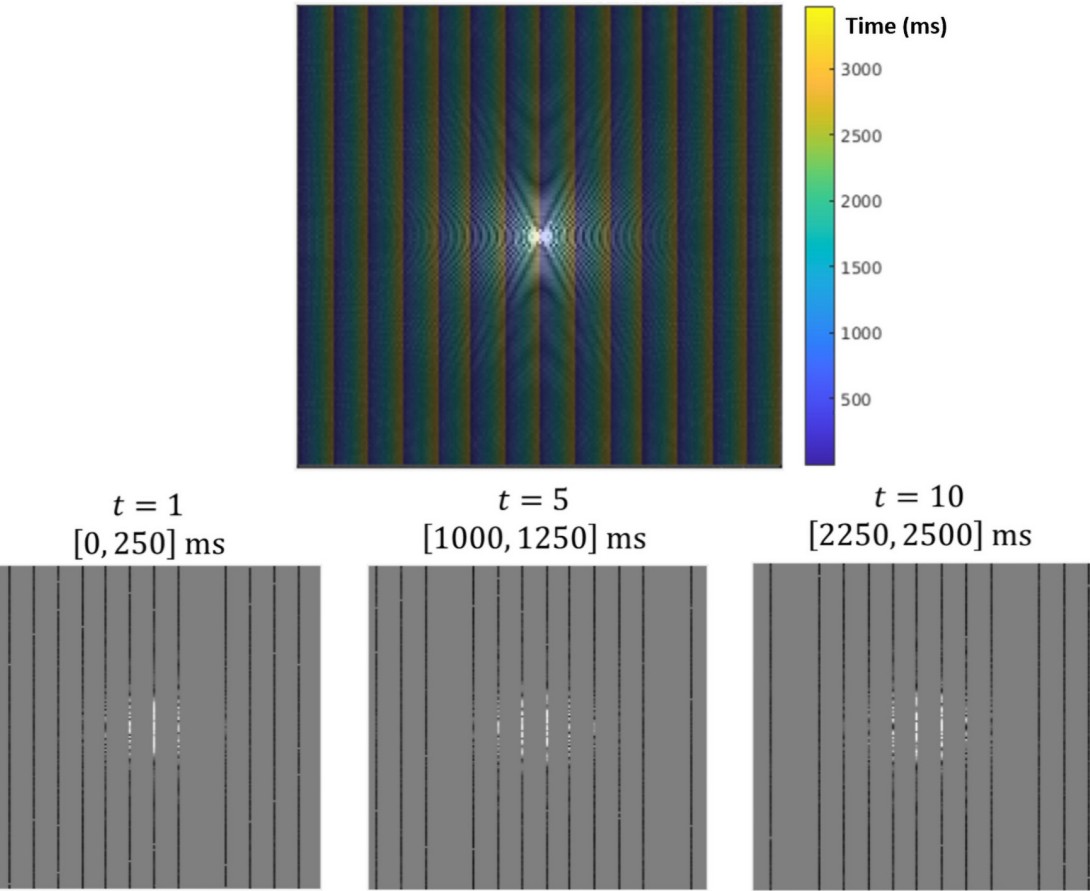

**Fig 5. An illustration of the $k$-space partitioning process.** (Top) The sampling scheme displayed in Fig 3 is overlaid on sample $k$-space data. All $k$-space measurements in the same time interval constrain the reconstruction of a single time-point in the accelerated reconstruction. (Bottom) These $k$-space points form the "measured" partition of the reconstructed dataset. The temporal resolution of the reconstruction determines the size of the measured partition of $k$-space points used for each reconstruction. The higher the acceleration factor $\alpha$, the shorter the duration of each measured partition and the more underdetermined the reconstruction problem.

a. A convex, quadratic smoothness penalty on $X = (X_1, \ldots, X_T)$ applied separately to each voxel $v = 1, \ldots, V$ in the spatial domain. When a voxel enhances smoothly, the penalty is small; when it doesn't, the penalty is large. The penalty can be weighted differently on each voxel $v$, reflecting spatial variation in desired enhancement smoothness.

b. A data fidelity constraint on $\tilde{X} = (\tilde{X}_1, \ldots, \tilde{X}_T)$, requiring that, for each $t$, any entry of $\tilde{X}_t$ measured during time interval $t$ must exactly match the data acquired during that interval.

3. Solve for the image $X = (X_1, \ldots, X_T)$ that minimizes the smoothness penalty and satisfies the data fidelity constraint on $\tilde{X} = (\tilde{X}_1, \ldots, \tilde{X}_T)$.

Intuitively, this optimization is a search for the smoothest set of enhancement curves that are consistent with our measured $k$-space data. Formulating the reconstruction in this way relies heavily on one key assumption: *enhancement is smooth on the timescale of the reconstruction's temporal resolution*. Requiring smoothness on short timescales does not limit the ability of this algorithm to accurately measure sharp spatio-temporal changes as in the AIF, since these changes occur on longer timescales. We chose a target temporal resolution of 0.25 seconds in response to speculations in the literature [12–16] about optimal temporal resolutions for pharmacokinetic analysis in breast DCE-MRI.

For a formal description of the partitioning process invoked above, see S1 Appendix. Since the smoothness penalty optimized during reconstruction is a positive-definite quadratic form, the reconstruction optimization is convex and has a unique solution. While this solution can be defined analytically, we calculate it iteratively via conjugate gradient descent. See S2 and S3 **Appendices** for further details. Finally, the computation of image updates requires regularization to converge; for a discussion on choice of regularization parameter, see S4 Appendix. Documentation and demos for the phantom, scanner, and reconstruction pipelines are available at <github.com/tyo8/ECA_Demo>.

## Data analysis

As an initial investigation of the ECA reconstruction framework, we compared images and enhancement curves recovered from ECA and standard IFFT reconstructions. Two parameters, bolus arrival time (BAT) and initial enhancement slope, were extracted from the signal enhancement curve of vessel and lesion voxels by ECA and standard IFFT methods.

BAT was measured from time of peak enhancement in vessel voxels. In lesion voxels, BAT was calculated as the earliest time at which voxels reached or exceeded 20% of their maximum enhancement over baseline.

To calculate initial slope in vessel voxels, each voxel timeseries was interpolated by a modified Akima method [47]. The initial slope was the maximum first derivative of the interpolated AIF curve. In lesion voxels, percent signal enhancement (PSE) versus time was fitted to a piecewise empirical mathematical model (EMM):

$$PSEt = \begin{cases} \dfrac{A * (\alpha(t - t_0))^2}{1 + (\alpha(t - t_0))^2}, & (t \geq t_0) \\[2ex] 0, & (t < t_0) \end{cases}$$

where $t_0$ is the BAT in lesion voxels, $A$ is the upper limit of percent enhancement, and $\alpha$ is the uptake rate; thus, $A\alpha$ is the initial enhancement slope.

To assess image preservation, voxel-wise image fidelity was also compared between the two methods. Ground-truth images are computed by evaluating the signal function at the center of the temporal window surrounding each reconstructed time point. The distribution of absolute voxel-wise signal differences between reconstructed and ground-truth images is then computed and summarized.

## Results

### Bolus arrival time (BAT)

Sample BAT maps, computed from both the IFFT and ECA reconstructions, are shown in Fig 6. The images computed from an ECA reconstruction show more accurate and precise bolus arrival time estimate than do the images computed from an IFFT reconstruction (Fig 7a and 7b). BAT estimation error distributions are shown (for lesion and vessel voxels) for all cases in Fig 7a and 7b.

ECA was substantially more accurate than IFFT in most cases for predicting lesion BAT and in all cases for predicting vessel BAT. If we treat each voxel-wise BAT measurement as an independent sample from all possible locations in the image, then we can treat set of ratios of measurement errors (ECA error)/(IFFT error) at each voxel as a distribution; this distribution quantifies the distribution between the two methods. Where this ratio (bounded by confidence intervals) is less than 1, ECA predicts BAT more accurately than IFFT; where the ratio is larger than 1, IFFT predicts BAT better than ECA. We report the median error ratio in lesions and in voxels over all cases, and we take a $5\sigma$ confidence interval (corresponding to $p < 10^{-6}$) about the median error ratio. In lesion voxels, we find a median error ratio of $0.210 \pm 0.013$, corresponding to a more than 4-fold BAT accuracy improvement in ECA; in vessel voxels, we find a median error ratio of $0.0825 \pm 0.0018$, corresponding to a more than 11-fold BAT accuracy increase in the ECA reconstruction. Summary statistics over all cases are shown in Table 3.

BAT differences are shown separated by case in Fig 8. ECA was substantially more accurate than IFFT in most cases for predicting lesion BAT and in all cases for predicting vessel BAT. (Table 3).

Images reconstructed from the ECA algorithm show much greater precision in estimation of bolus arrival time. Errors in BAT have much smaller spread (median absolute deviation) for ECA reconstructions than for IFFT reconstructions, especially in vessel voxels. Furthermore, the BAT error distribution is clustered much nearer to 0 in ECA reconstructions than in IFFT reconstructions (Fig 7a and 7b), especially in vessel voxels.

Because bolus arrival time estimates were more accurate with ECA, we conclude that ECA reconstruction allows for more accurate and more precise bolus tracking than traditional

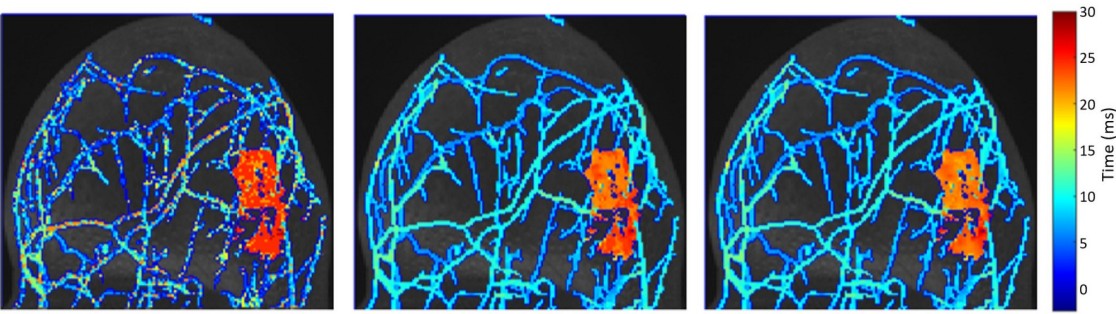

**Fig 6. Bolus arrival time computed from the case 4 image set.** From left to right: IFFT reconstruction, ECA reconstruction, and ground truth. Times shown on the color bar are measured in seconds.

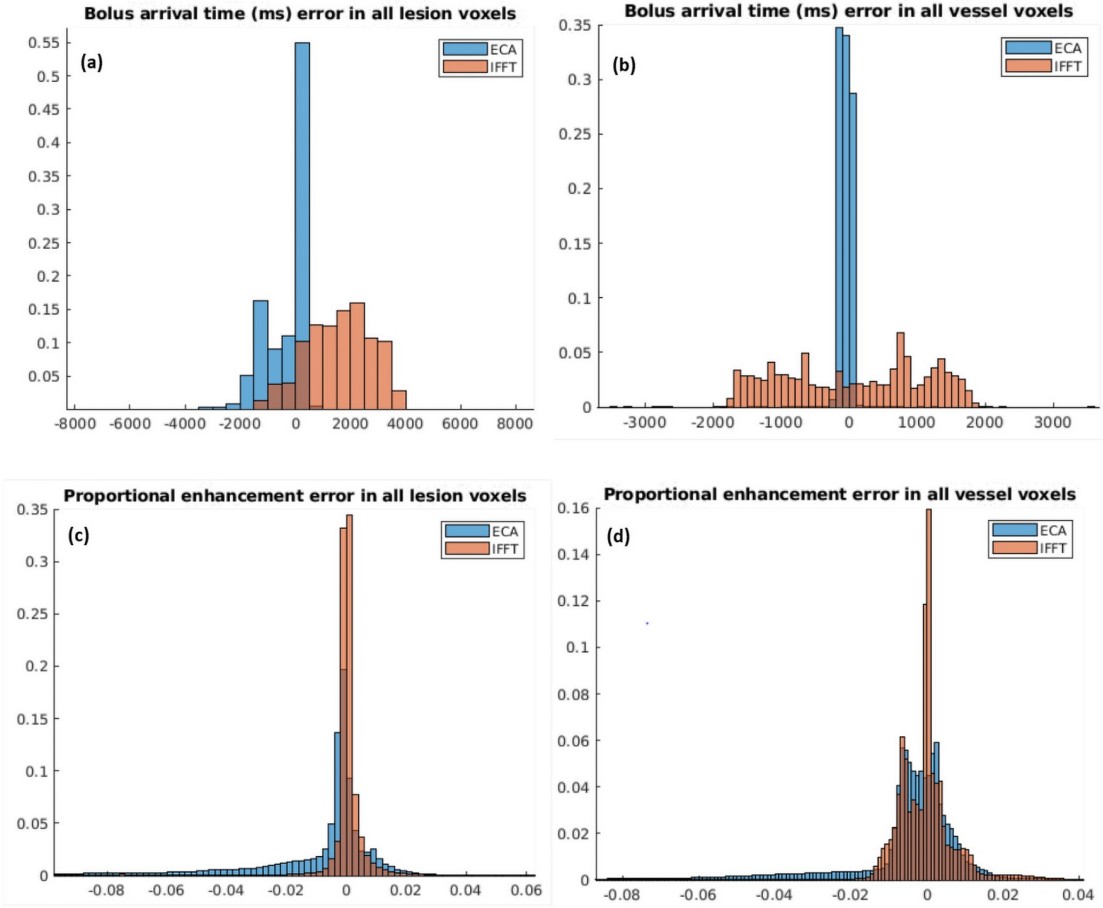

**Fig 7. Error distributions are shown for all cases.** Distributions for IFFT and ECA are shown in different colors on the same plot. (a) and (b) show errors in the estimation of the bolus arrival time in milliseconds; (c) and (d) show the distribution of the proportional voxel error.

**Table 3. Summary statistics for BAT and voxel intensity errors in ECA and IFFT reconstructions.**

| | | **BAT Paired Error Ratio (ECA error/IFFT error)** | | | | | |
|---|---|---|---|---|---|---|---|
| | | Case 1 | Case 2 | Case 3 | Case 4 | Case 5 | All Cases |
| **Lesions** | | 0.0803 ± 0.0049 | 1.46 ± 0.24 | 0.282 ± 0.042 | 0.156 ± 0.009 | 1.21 ± 0.12 | 0.210 ± 0.013 |
| **Vessels** | | 0.0688 ± 0.0026 | 0.085 ± 0.006 | 0.1049 ± 0.0059 | 0.0695 ± 0.0038 | 0.1020 ± 0.0044 | 0.0825 ± 0.0018 |
| | | **Voxel Error (%)** | | | | | |
| | | **Case 1** | **Case 2** | **Case 3** | **Case 4** | **Case 5** | **Overall** |
| **Lesion** | **IFFT** | 0.079 | 0.003 | 0.12 | 0.058 | 0.054 | 0.007 |
| | **ECA** | 0.56 | 0.40 | 0.51 | 0.53 | 0.27 | 0.47 |
| **Vessel** | **IFFT** | 0.39 | 0.35 | 0.42 | 0.40 | 0.39 | 0.39 |
| | **ECA** | 0.49 | 0.56 | 0.66 | 0.57 | 0.59 | 0.56 |

For BAT error, the median paired error ratio is shown with a 99% confidence interval. Median absolute deviation is reported for voxel intensity error.

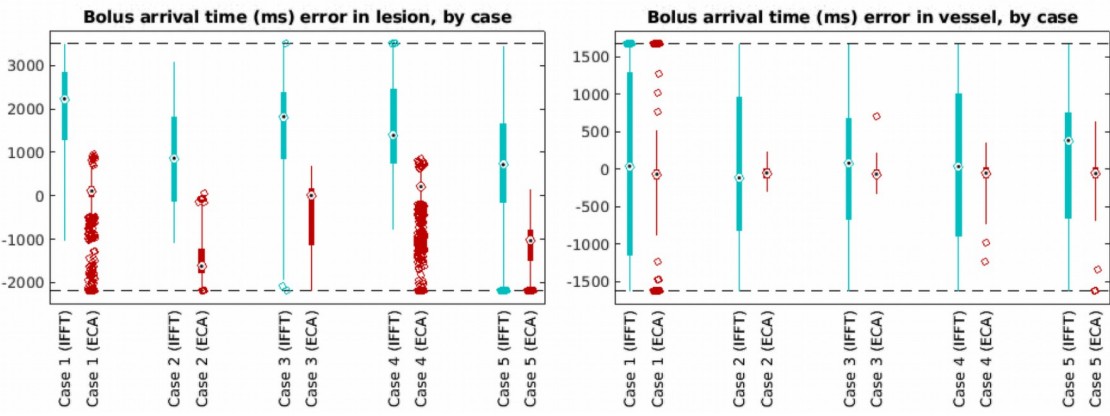

**Fig 8. Box and whisker plots of the error in bolus arrival time, by case number and reconstruction method.** Red boxplots (right) are from ECA-reconstructed data; blue boxplots (left) are from standard IFFT reconstructions. Separate sets of plots are shown for lesion and vessel voxels.

ultrafast methods. This increase in temporal precision comes at the cost of a small decrease in image-intensity preservation; however, the errors in voxel intensity remain very small in both cases.

## Initial slope

Enhancement curves recovered from reconstructed images closely match simulated enhancement from the phantoms. Fig 9 plots ground truth versus estimate values for the initial slope, as derived from both ECA (left) and IFFT (right) reconstructions. Compared to standard IFFT, ECA reconstruction more accurately recovers the initial slope of the enhancement curve in both vessel and lesion voxels. To see this, first note that the coefficient of determination in both sets of truth-estimate fits is larger for ECA than IFFT; therefore, ECA produces lower-variance estimates of initial slope than IFFT does. Next, compare the slopes and offsets of the truth-estimate fits. In vessel voxels, IFFT and ECA have similar fit slopes and offsets, and therefore introduce similar amounts of bias; in lesion voxels, ECA introduces much less bias than IFFT. Although both methods exhibit greater error when recovering enhancement curves with larger initial enhancement slope, ECA estimates the initial slope more accurately than standard IFFT.

## Image fidelity

ECA-reconstructed images are highly similar to ground-truth images. A sample error map is overlaid on a phantom in Fig 10. Voxel-wise error statistic summaries are shown in Fig 11 for the 5 phantoms tested, and a sample enhancing-voxel error distribution is shown in Fig 7c and 7d. Voxel-wise errors in the IFFT reconstruction were generally smaller than in the ECA reconstruction, though fidelity errors were very small in both methods. The voxel intensity error distributions shown in Fig 10 show that the increased temporal resolution comes with at most a negligible cost in voxel intensity accuracy.

## Comparison with standard methods

Fig 12 juxtaposes a median-quality curve from the ECA reconstruction with the IFFT reconstruction (which assumes perfect SENSE and Partial Fourier reconstructions) of the same voxel. Sample curves are shown for constant, vessel, and lesion voxels.Note the "undershoot"

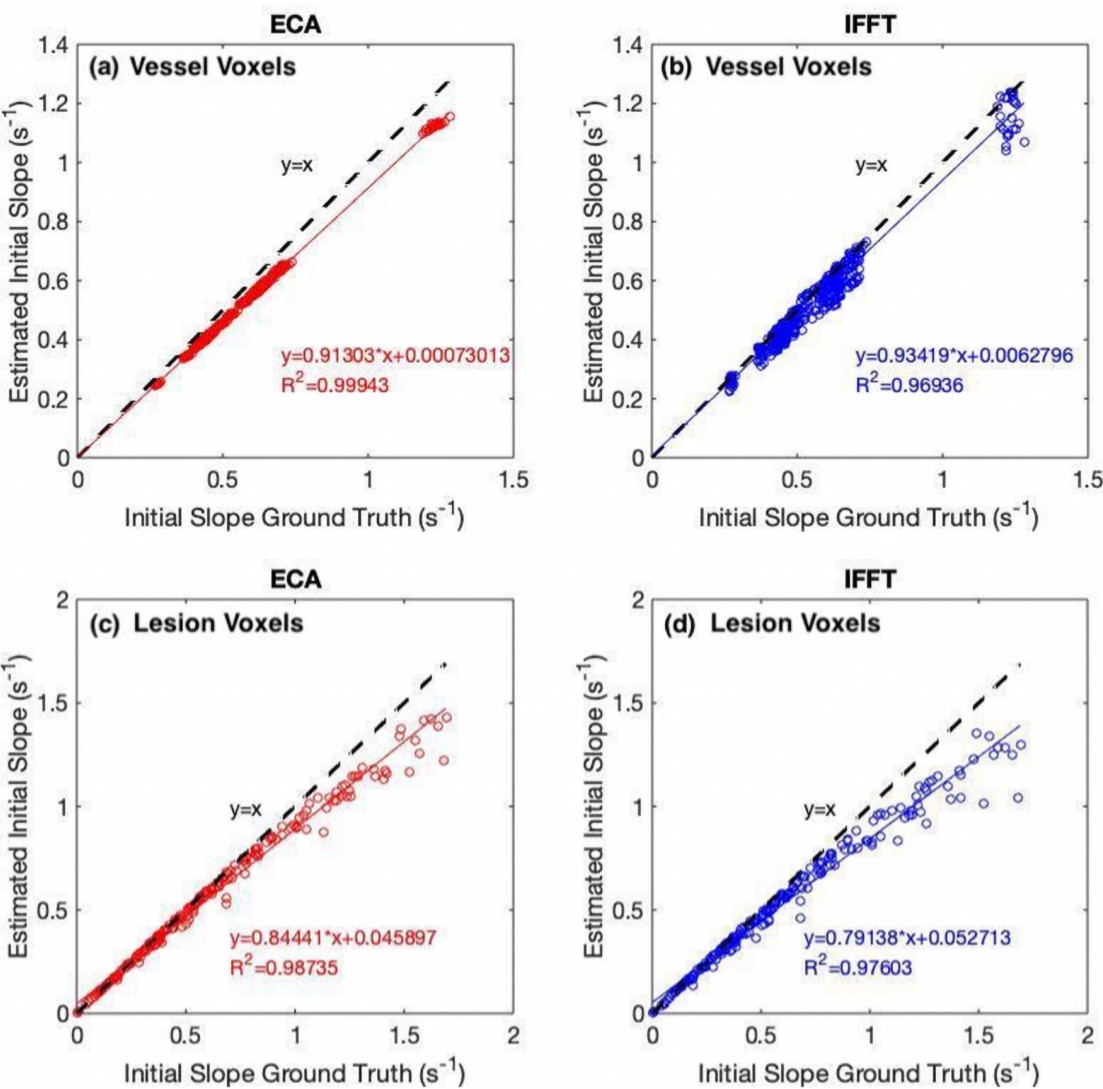

**Fig 9. Scatter plot between ground truth initial slope and estimated initial slope.** The panels show ECA and standard IFFT in (a) (b) vessels voxels and (c)(d) lesion voxels. The red lines and blue lines represent the linear correlations and black dashed lines show unity.

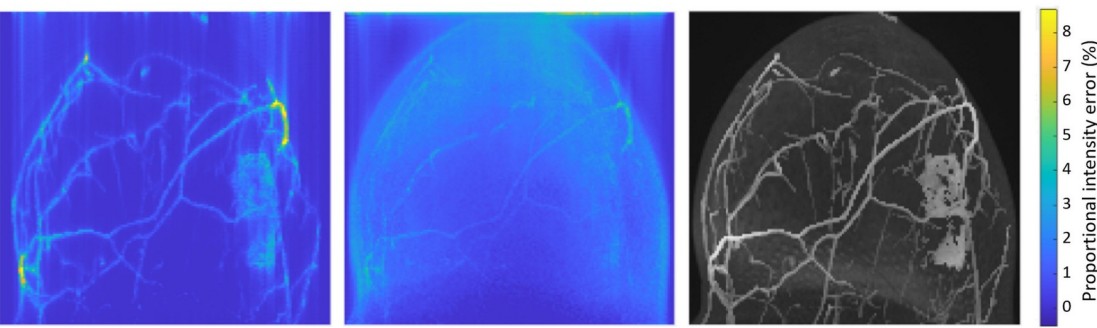

**Fig 10. Proportional intensity error per voxel for case 4.** Proportional intensity error is shown from the mean projection over time and through the volume. From left to right: IFFT reconstruction error, ECA reconstruction error, and the ground truth image.

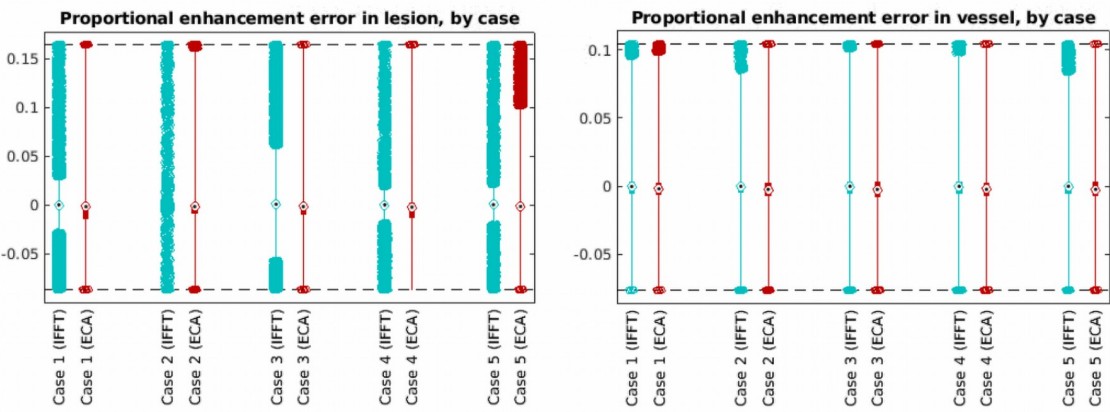

**Fig 11. Box and whisker plots of the proportional enhancement error, by case and reconstruction method.** Red boxplots (right) are from ECA-reconstructed data; blue boxplots (left) are from standard IFFT reconstructions. Separate plots are shown for lesion and vessel voxels.

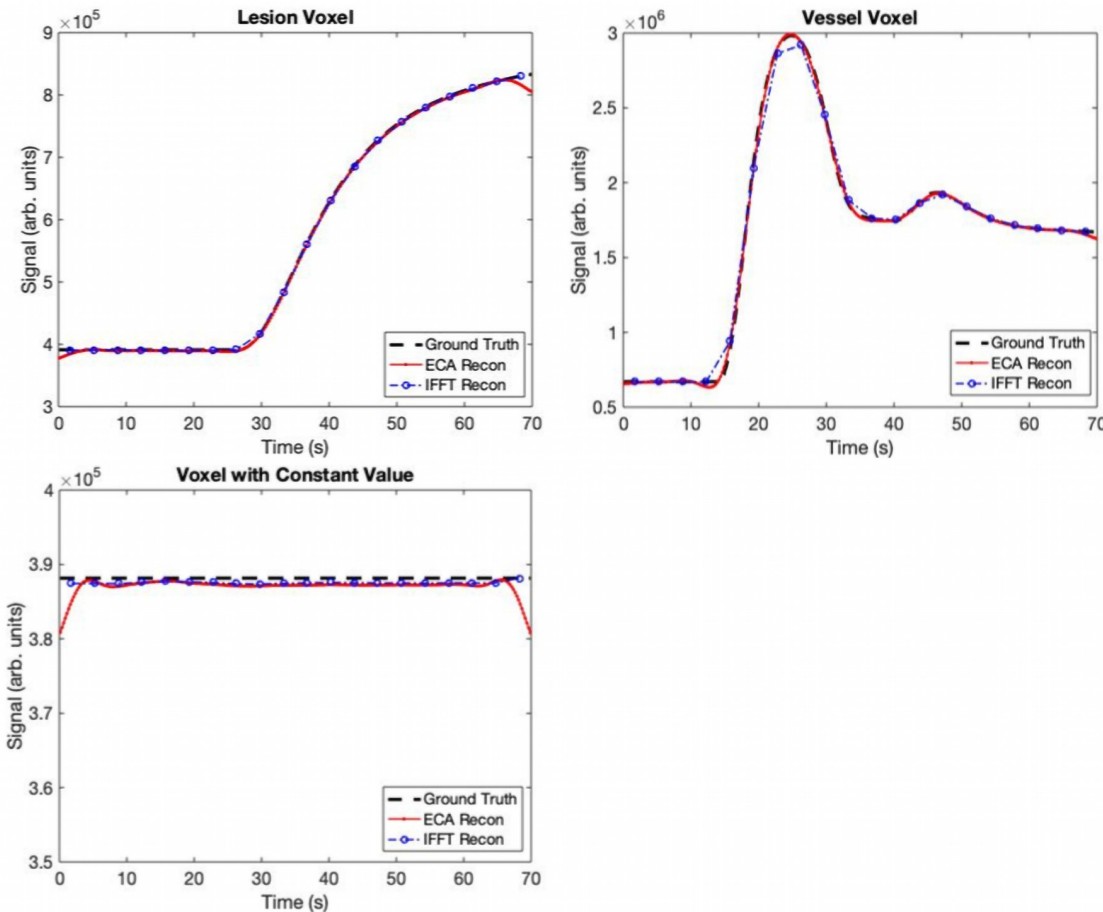

**Fig 12. Comparison of sample signal curves from standard IFFT and ECA reconstruction.** The ECA is more sensitive to noise than the IFFT, but the noise is still small with respect to the signal. IFFT reconstruction estimates bolus arrival time and peak signal less accurately than ECA reconstruction.

of the ECA reconstruction at the first and last time points of the reconstructed curve; it is most visible in the constant curve, though small compared to signal intensity values in all cases. While they are deviations from the true signal curve, they do not pose serious impediments to pharmacokinetic analysis using ECA. This is true first and foremost because we can easily remove the first and last time points when calculating kinetic parameters. In addition, the "undershoot" of the curve at the first point is a straightforward product of the fact that (1) we initialize by zero-filling and (2) low-frequency components of $k$-space have only been sparsely sampled within the first 250 ms of scanning. Initializing the reconstruction with a static pre-contrast image would likely remove this problem and is under investigation as a future strategy. The undershoot at the last time point is a problem of extrapolation; our method is closely related to splines, which interpolate very well but tend to extrapolate poorly beyond the data domain.

Overall, the ECA reconstruction captured bolus arrival times in enhancing voxels more accurately than the IFFT reconstruction, suffering only a small loss of accuracy in estimating the per-time point image (Table 3 and **Figs** 7c and 7d and 11). While ECA proved uniformly more accurate in recovering the BAT in the vessel, ECA and IFFT estimated lesion BAT with similar bias in two cases; in the other three, ECA estimated the BAT with lower bias and variance. Even in cases where ECA and IFFT produced similarly biased estimations of BAT, the ECA estimated the BAT with lower variance (Table 3 and **Figs** 7a and 7b and 8).

## Discussion

The results from realistic phantoms reported here demonstrate that sparse uniform samples of $k$-space can be used to reconstruct DCE-MRI breast images with high fidelity and very high temporal resolution. This allows significantly more accurate arterial bolus tracking and more accurate measurement of lesion enhancement parameters such as the bolus arrival time and initial enhancement slope. These important diagnostic parameters have been used to improve cancer diagnosis [5,12,48,49]. Since the early phase of enhancement is critical for distinguishing cancers from background parenchymal enhancement [7,50], high fidelity high temporal resolution images produced with ECA may significantly improve identification and characterization of small cancers.

The ECA method introduced here is based on two primary principles.

1. If $k$-space data is partitioned into small subsets by acquisition time, each subset retains important kinetic information.
   Especially when enhancement is sparse (as in the early phase of contrast uptake), even highly sub-Nyquist acquisitions contain sufficient information to almost fully constrain the evolution of contrast kinetics. The UnWRAP sequence used in this study demonstrates this principle in action, using simple uniform undersampling to sample a representative bandwidth of spatial frequencies. We believe the UnWRAP $k$-space ordering scheme to be a good choice for sampling the early phase of contrast uptake, but we emphasize that this principle is applicable to any known/deterministic $k$-space sampling trajectory.

2. DCE-MRI enhancement is approximately smooth in small time intervals.
   Provided kinetic processes are slow on fractional-second timescales and samples are acquired with sufficient SNR and bandwidth, very few measurements of $k$-space are needed to "tie together" the time-evolution of an image set. This is especially true when all of the partial $k$-space measurements taken together form a Nyquist-complete set. We designed the UnWRAP acquisition sequence to maximally leverage this principle, but it is applicable to many undersampled reconstruction methods in DCE-MRI.

The UnWRAP method introduced here maintains relatively high SNR over each subset of *k*-space by sampling a mixture of high and low spatial frequencies. The extent to which both reconstruction methods preserved voxel-wise intensity suggests the UnWRAP *k*-space trajectory chosen offers some advantages over a standard sequential acquisition. Because it maintains a uniform frequency density in the scan, the UnWRAP sequence samples the *k*-space center often enough to preserve signal intensity and the *k*-space edges often enough to correctly assign signal to spatial features. As is true for any acceleration method, effective application of the UnWRAP method requires adequate SNR during each measurement interval.

In addition, we comment on some of the assumptions made in this simulation study with respect to parallel imaging. While the current simulation assumed that the sweep time was the temporal resolution of the original images (3.5 s), in this proof-of-principle study we did not take into account that these images were acquired with parallel imaging (SENSE). However, we do not believe the ECA method to be incompatible with parallel imaging. While ECA does impose constraints on spatial data by applying masking in k-space, the reconstruction optimizes only in the temporal domain. Since SENSE spatially optimizes image data for compatibility with coil sensitivity maps, the two approaches optimize with respect to non-overlapping sources of information and thus could be combined.

The results summarized here demonstrate that ECA combined with UNWRAP sampling has promise for improving breast cancer screening and diagnosis. However, this study had some limitations:

- This was a simulation study, and it will be critical to test these results *in vivo*. These tests are currently underway.

- Motion artifacts were not included in this work. It will be critical to evaluate effects of motion in future simulations as well as in *in vivo* studies.

- Neither heart nor background enhancement were modeled in these simulations. It will be critical to assess the capacity of ECA to reconstruct diagnostically useful enhancement in the presence of background enhancement.

- T2$^*$ effects were not simulated. These effects are significant during the early phase of contrast media uptake, especially in arteries.

- Other sampling trajectories were not tested; because not all scanners can implement all undersampling trajectories, it will be important to test ECAs performance with other types of accelerated acquisitions.

In addition to addressing the study limitations listed above, we suggest several further avenues of future investigation. First and foremost, ECA requires a thorough characterization of its performance across a wide range of noise levels. Second, because of the ubiquity of partial Fourier, parallel imaging, and ML acceleration methods, we will integrate our acceleration algorithm with popular implementations of these. Finally, we hope to test ECA on a wide variety of sampling trajectories and use this process to evaluate the optimality of both ECA and these trajectories in a wider context of DCE-MRI acceleration methods.

Taken as a whole, the data presented in this work constitute an argument that "sparse + smooth enhancement" characterize contrast kinetics in breast DCE-MRI to very high precision during the early phase of contrast media uptake. Smooth enhancement is a stringent condition to impose on DCE-MRI data and, on its own, encodes a great deal of physiological structure. Within such a constraint, even a small number of well-chosen measurements can closely characterize early enhancement in the breast. ECA reconstruction provides a robust

framework to increase diagnostic accuracy and improve understanding of hemodynamics in normal breast and cancers.

## Supporting information

**S1 Fig. Reconstruction error and convergence speed as a function of regularization.** (Blue) Error, measured here by normalized mean-square error (MSE), increases with regularization strength. (Red) Computation time, measured in number of iterations, decreases with regularization strength.
(TIF)

**S1 Appendix. Formal description of partition constraints.**
(DOCX)

**S2 Appendix. The penalty function.**
(DOCX)

**S3 Appendix. Solutions to the optimization problem.**
(DOCX)

**S4 Appendix. Regularization in the optimization problem.**
(DOCX)

## Acknowledgments

The authors would like to thank Michael Bian for feedback on an earlier draft of this manuscript.

## Author Contributions

**Conceptualization:** Ty O. Easley, Byol Kim, Gregory S. Karczmar, Rina F. Barber, Federico D. Pineda.

**Data curation:** Ty O. Easley, Byol Kim, Gregory S. Karczmar, Federico D. Pineda.

**Formal analysis:** Ty O. Easley, Zhen Ren, Byol Kim, Rina F. Barber, Federico D. Pineda.

**Funding acquisition:** Gregory S. Karczmar, Rina F. Barber.

**Investigation:** Ty O. Easley, Zhen Ren, Byol Kim, Gregory S. Karczmar, Rina F. Barber, Federico D. Pineda.

**Methodology:** Ty O. Easley, Byol Kim, Gregory S. Karczmar, Rina F. Barber, Federico D. Pineda.

**Project administration:** Ty O. Easley, Gregory S. Karczmar, Rina F. Barber, Federico D. Pineda.

**Resources:** Gregory S. Karczmar.

**Software:** Ty O. Easley, Zhen Ren, Byol Kim, Rina F. Barber.

**Supervision:** Ty O. Easley, Byol Kim, Gregory S. Karczmar, Rina F. Barber, Federico D. Pineda.

**Validation:** Ty O. Easley, Zhen Ren, Byol Kim, Gregory S. Karczmar, Rina F. Barber, Federico D. Pineda.

**Visualization:** Ty O. Easley, Rina F. Barber, Federico D. Pineda.

**Writing – original draft:** Ty O. Easley, Zhen Ren, Rina F. Barber.

**Writing – review & editing:** Ty O. Easley, Zhen Ren, Byol Kim, Gregory S. Karczmar, Rina F. Barber, Federico D. Pineda.

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
