## [Decision Letter · Decision Letter 0]

10 Feb 2021

PONE-D-20-33045

Enhancement-Constrained Acceleration: A Robust Reconstruction Framework in Breast DCE-MRI

PLOS ONE

Dear Dr. Easley,

Thank you for submitting your manuscript to PLOS ONE. After careful consideration, we feel that it has merit but does not fully meet PLOS ONE’s publication criteria as it currently stands. Therefore, we invite you to submit a revised version of the manuscript that addresses the points raised during the review process.

We look forward to receiving your revised manuscript.

Kind regards,

Xi Chen

Academic Editor

PLOS ONE

Journal Requirements:

Reviewers' comments:

Reviewer's Responses to Questions

**Comments to the Author**

1. Is the manuscript technically sound, and do the data support the conclusions?

Reviewer #1: Yes

Reviewer #2: Yes

2. Has the statistical analysis been performed appropriately and rigorously? 

Reviewer #1: Yes

Reviewer #2: Yes

3. Have the authors made all data underlying the findings in their manuscript fully available?

Reviewer #1: Yes

Reviewer #2: Yes

4. Is the manuscript presented in an intelligible fashion and written in standard English?

Reviewer #1: Yes

Reviewer #2: Yes

5. Review Comments to the Author

Reviewer #1: Comments to the Authors:

This study proposed an innovative framework of k-space sampling trajectory and reconstruction to accelerate breast DCE-MRI, which is essential for improving the ability of DCE-MRI to capture early-phase contrast agent kinetics. A very impressive reconstructed temporal resolution of 0.25 s was achieved by this method, without significantly sacrificing spatial resolution or SNR. The manuscript is well-written, with a clear description of the proposed algorithms and evaluation. In summary, I recommend publishing of this manuscript with some minor refinements. Please see specific comments below.

Specific Comments:

1. In “Theory and Methods – Virtual Phantoms” section:

Please provide the specific models / equations and values of associated parameters used to produce the concentration curves in vessels and lesions.

2. In “Theory and Methods – Virtual Scanner” section:

In the simulation presented in this study, path times were computed to match standard ultrafast time resolutions (i.e., sweep time = 3.5s). However, in real practice, to reach this sweep time parallel imaging would be necessary. Please address how the consideration of parallel imaging implementation would change the proposed method in the Discussion session.

3. In “Results – Bolus Arrival Time (BAT)”:

line 357 --

Please correct “IFT” to “IFFT”.

4. In “Results – Bolus Arrival Time (BAT)”:

lines 358 - 359 --

The comments in this part are confusing. Is BAT estimated with ECA better than the one estimated with IFFT in cases 1, 3, and 4, or is BAT estimated with ECA in cases 1, 3, and 4 better than BAT estimated with ECA in cases 2 and 5? Does the substantial difference show statistical significance?

5. In Figure 10:

Is the unit of colorbar percentage?

6. In Figure 12:

Please comment on the observation that ECA curves show “undershoots” at the first and last time points. Is it due to extrapolation? How is it going to affect the evaluation and kinetic analysis?

Reviewer #2: The work demonstrates a new reconstruction method for semi-qualitative breast DCE-MRI data. In particular, the authors claim that the method provides high fidelity reconstruction at 0.25s temporal resolution. The work develops a virtual scanner which is used for simulating k-space data and subsequent sampling. This is a useful contribution. Also, the code has been shared on Github useful for reproducible research. However, I have some concerns with the data simulations, assumptions and claims (see minor comments related to opinions) which are detailed below. Detailed comments attached in file.

6. PLOS authors have the option to publish the peer review history of their article (what does this mean?). If published, this will include your full peer review and any attached files.

Reviewer #1: No

Reviewer #2: **Yes: **Sairam Geethanath

---

## [Author Response · Author response to Decision Letter 0]

10 Aug 2021

We have attached a reviewer response letter elsewhere in the submission.

---

## [Decision Letter · Decision Letter 1]

4 Oct 2021

Enhancement-Constrained Acceleration: A Robust Reconstruction Framework in Breast DCE-MRI

PONE-D-20-33045R1

Dear Dr. Easley,

We’re pleased to inform you that your manuscript has been judged scientifically suitable for publication and will be formally accepted for publication once it meets all outstanding technical requirements.

Kind regards,

Xi Chen

Academic Editor

PLOS ONE

Additional Editor Comments (optional):

Reviewers' comments:

Reviewer's Responses to Questions

**Comments to the Author**

1. If the authors have adequately addressed your comments raised in a previous round of review and you feel that this manuscript is now acceptable for publication, you may indicate that here to bypass the “Comments to the Author” section, enter your conflict of interest statement in the “Confidential to Editor” section, and submit your "Accept" recommendation.

Reviewer #1: All comments have been addressed

Reviewer #3: All comments have been addressed

2. Is the manuscript technically sound, and do the data support the conclusions?

Reviewer #1: Yes

Reviewer #3: Yes

3. Has the statistical analysis been performed appropriately and rigorously? 

Reviewer #1: Yes

Reviewer #3: Yes

4. Have the authors made all data underlying the findings in their manuscript fully available?

Reviewer #1: Yes

Reviewer #3: Yes

5. Is the manuscript presented in an intelligible fashion and written in standard English?

Reviewer #1: Yes

Reviewer #3: Yes

6. Review Comments to the Author

Reviewer #1: (No Response)

Reviewer #3: (No Response)

7. PLOS authors have the option to publish the peer review history of their article (what does this mean?). If published, this will include your full peer review and any attached files.

Reviewer #1: No

Reviewer #3: No

---

## [Editor Report · Acceptance letter]

13 Oct 2021

PONE-D-20-33045R1 

Enhancement-Constrained Acceleration: A Robust Reconstruction Framework in Breast DCE-MRI 

Dear Dr. Easley:

I'm pleased to inform you that your manuscript has been deemed suitable for publication in PLOS ONE. Congratulations! Your manuscript is now with our production department. 

Kind regards, 

on behalf of

Dr. Xi Chen 

Academic Editor

PLOS ONE